# Molecular characterization of *Salmonella* spp. and *Listeria monocytogenes* strains from biofilms in cattle and poultry slaughterhouses located in the federal District and State of Goiás, Brazil

Emilia Fernanda Agostinho Davanzo[1]*, Rebecca Lavarini dos Santos[1‡], Virgilio Hipólito de Lemos Castro[1‡], Joana Marchesini Palma[1‡], Bruno Rocha Pribul[2‡], Bruno Stéfano Lima Dallago[1‡], Bruna Fuga[3‡], Margareti Medeiros[1‡], Simoneide Souza Titze de Almeida[1‡], Hayanna Maria Boaventura da Costa[1‡], Dália dos Prazeres Rodrigues[2‡], Nilton Lincopan[3‡], Simone Perecmanis[1‡], Angela Patrícia Santana[1]

1 Faculty of Agronomy and Veterinary Medicine, University of Brasília (UnB), Brasília, DF, Brazil, 2 National Reference Laboratory for Bacterial Enteric Infections, Oswaldo Cruz Institute, Manguinhos, Rio de Janeiro, RJ, Brazil, 3 Laboratory of Bacterial Resistance and Therapeutic Alternatives, Biomedical Sciences Institute, University of São Paulo, São Paulo, SP, Brazil

☯ These authors contributed equally to this work.
‡ These authors also contributed equally to this work.
* emiliadavanzo@unb.br

**Data Availability Statement:** All relevant data are within the manuscript and its Supporting Information files.

## Abstract

*Listeria monocytogenes* and *Salmonella* spp. are considered important foodborne pathogens that are commonly associated with foods of animal origin. The aim of this study was to perform molecular characterization of *L. monocytogenes* and *Salmonella* spp. isolated from biofilms of cattle and poultry slaughterhouses located in the Federal District and State of Goiás, Brazil. Fourteen *L. monocytogenes* isolates and one *Salmonella* sp. were detected in poultry slaughterhouses. No isolates were detected in cattle slaughterhouses. All *L. monocytogenes* isolates belonged to lineage II, and 11 different pulsotypes were detected. Pulsed-field gel electrophoresis analysis revealed the dissemination of two strains within one plant, in addition to the regional dissemination of one of them. The *Salmonella* isolate was identified via whole genome sequencing as *Salmonella enterica* serovar Minnesota ST548. In the sequence analysis, no premature stop codons were detected in the *inlA* gene of *Listeria*. All isolates demonstrated the ability to adhere to Caco-2 cells, while 50% were capable of invading them. Antimicrobial resistance was detected in 57.1% of the *L. monocytogenes* isolates, and resistance to sulfonamide was the most common feature. The *tetC*, *ermB*, and *tetM* genes were detected, and four isolates were classified as multidrug-resistant. *Salmonella* sp. was resistant to nine antimicrobials and was classified as multidrug-resistant. Resistance genes *qnrB19*, *bla*$_{CMY-2}$, *aac(6')-Iaa*, *sul2*, and *tetA*, and a mutation in the *parC* gene were detected. The majority (78.5%) of the *L. monocytogenes* isolates were capable of forming biofilms after incubation at 37°C for 24 h, and 64.3% were capable of forming biofilms after incubation at 12°C for 168 h. There was no statistical difference in the

**Funding:** APS was supported by FAPDF (spontaneous demand n.3/2016).EFAD was funded as well by CAPES, via PROAP and DPG of University of Brasília. The funders had no role in study design, data collection and analysis, decision to publish, or preparation of the manuscript.

**Competing interests:** The authors have declared that no competing interests exist.

biofilm-forming capacity under the different evaluated conditions. *Salmonella* sp. was capable of forming biofilms at both tested temperatures. Biofilm characterization was confirmed by collecting the samples consistently, at the same sampling points, and by assessing biofilm formation *in vitro*. These results highlight the potential risk of cross-contamination in poultry slaughterhouses and the importance of surveillance and pathogen control maintenance programs within the meat production industry.

## Introduction

The hygienic and sanitary conditions of the industrial environment are essential for ensuring food safety and quality [1–3]. One of the current challenges encountered in slaughtering and meat processing is the prevention of cross-contamination by microorganisms [4, 5]. Cross-contamination can occur via contact with bacterial biofilms due to the performance of improper hygiene and sanitization practices at the slaughter system. This leads to a decrease in food quality, resulting in pathogen transmission and the occurrence of recurring contamination of food [5, 6]. *Listeria monocytogenes* and *Salmonella* spp. are deemed important pathogens whose dissemination can be controlled to ensure food quality and safety. Their presence is a criterion for evaluation within the various aspects of quality control of the meat production process [7].

*L. monocytogenes* is recognized as the main etiological agent of listeriosis in humans [8]. It is a psychrophilic bacterium, that can develop on surfaces of meat production industries [9]. The species is subdivided into 13 serotypes, of which 1/2a, 1/2c, and 4b are the most commonly reported serotypes involved in cases of human listeriosis [10]. Owing to the difficulties and limitations encountered with methods of traditional serology, different subtyping methods have been developed, such as the differentiation of the main lineages assessed via polymerase chain reaction (PCR) [11]. Additionally, molecular typing methods with high discriminatory power, such as pulsed-field gel electrophoresis (PFGE), are widely used to assess clonal dissemination within industries [10, 12, 13]. There is considerable pathogenic potential among the different *L. monocytogenes* clones. [14]. The role of internalin A in *Listeria*, encoded by the *inlA* gene, is exemplary as its functionality is necessary for pathogen internalization [15–17]. The presence of premature stop codons (PMSCs) in this gene has been associated with clones with an attenuated invasion phenotype [18].

The genus *Salmonella*, which is responsible for salmonellosis, includes the *Salmonella enterica* species. It is a microorganism that is often identified as an etiological agent of food outbreaks, presenting a complex epidemiology regarding its transmission and distribution, and is therefore important for investigation in the assessment of risks to public health. [19]. The species demonstrates adherence to surfaces such as metals, glass and rubber, and exhibits the ability of biofilm formation [20–22].

Antimicrobial resistance (AMR) is considered a grave concern and risk to human health [23, 24]. To mitigate its risks, the World Health Organization established a global action plan [25]. One of the main strategies outlined in the plan is to monitor the spread of AMR. Considering the importance of monitoring *L. monocytogenes* and *Salmonella* spp. dissemination and the necessity of ensuring public health and safety, and owing to the limited investigations related to the presence of biofilms in Brazil, the objectives of this study were to detect biofilm-induced contamination points and the presence of these microorganisms in processing plants of cattle and poultry slaughterhouses, to characterize these strains by performing serotyping,

AMR analysis, and to evaluate the *in vitro* biofilm formation capability of all isolates. Additionally, whole genome sequencing (WGS) of *Salmonella* spp. was performed in addition to the conduction of PFGE, sequencing of the *inlA* gene, and cell adhesion and invasion assays in Caco-2 cells for all *Listeria* strains.

## Material and methods

### Sample collection

Samples were collected from three poultry slaughterhouses (A, B and C) and two cattle slaughterhouses (D and E) located in the Federal District area (A, B and D) and in the State of Goiás (C and E). All slaughterhouses availed an official inspection service. All slaughterhouses agreed to participate voluntarily in the study. Sterile swabs (Absorve®, Brazil) and 25 cm$^2$ molds were used to examine the surfaces of the installations, equipment, and utensils in cattle and poultry slaughterhouses [26]. The sampling locations were separated into the following two groups: installations (floors, walls, and drains of dirty and clean areas), and equipment (mats, evisceration tables, chutes, and different machinery), and this method was based on protocols described by Barros *et al.* 2007 [27] and Nicolau and Bolocan 2014 [26]. Sixteen visits (eight in cattle slaughterhouses and eight in poultry slaughterhouses) were conducted between March 2017 and September 2018, for the collection of 287 swabs. Of the swabs collected, 118 were obtained from cattle slaughterhouses and 169 were obtained from poultry slaughterhouses. The swabs were individually added to tubes containing a sterile transport solution (peptone water 0.1%; Himedia®, India) and transported in isothermal boxes to the Food Microbiology Laboratory at the University of Brasília for culturing and microbiological isolation. Sample processing was performed within 24 h of sample collection. Biofilms were characterized within a processing plant by repetitively collecting samples at the same locations within each plant but on different visits, to observe the recurrence of the investigated microorganisms [28], and by assessing their *in vitro* biofilm-forming capability. Studies have correlated the permanence of microorganisms on surfaces after subjection to cleaning and sanitization practices to the existence of biofilms on such surfaces [29, 30]. Thus, sample collections were conducted at different times, with the surfaces treated hygienically and sanitized between visits.

### Isolation of *L. monocytogenes* and *Salmonella* spp

For the isolation of *L. monocytogenes*, swab samples were analyzed according to the methodology described by Ryser and Donnely 2015 [31]. Transport tubes containing swabs were homogenized in a tube shaker, with the subsequent addition of 1 mL of transport solution to 9 mL of UVM broth (Acumedia®, USA). After 24 h of incubation at 35˚C, the culture (0.1 mL) was transferred to 9 mL of Fraser broth (Acumedia®, USA). Esculin hydrolysis was performed after incubation at 35˚C for 24–48 h. Each sample was plated on Modified Oxoid (MOX) agar (Difco®, France). Presumptive colonies were incubated in Brain Heart Infusion (BHI) broth (Acumedia®, -USA) at 35˚C for 24 h and were biochemically analyzed. Isolates with characteristics compatible with the selected genus were examined by PCR to confirm the species using the primers LIP1 [32] and LIP2A [33] for amplification of the *prfA* gene. PCR was performed as per methods described by Kérouanton *et al.* 2010 [33]. Positive controls were provided by Dr. Ernesto Hofer of the Oswaldo Cruz Foundation located in Rio de Janeiro.

To identify species of *Salmonella*, swab samples were analyzed as per protocols described by Ryser and Donnely 2015 [31] and ISO 6579/2002 [34]. After homogenization of transport tubes containing the collected swab samples, each homogenate was individually transferred to tubes containing 1% buffered peptone water (Acumedia®, USA). Samples were incubated at

37˚C for 24 h. Aliquots were transferred to tetrathionate broth (Merck®, Germany), selenite cystine broth (Merck®, Germany), and Rappaport-Vassiliadis broth (Merck®, Germany). After incubation in selective media, samples were plated on Brilliant Green Phenol Red Agar (Acumedia®, USA), Xylose Lysine Deoxycholate (Merck®, Germany), and Bismuth Sulfite agars (Acumedia®, USA). Presumptive colonies were cultured on Nutrient agar (Acumedia®, USA) and were biochemically analyzed using appropriate tests [34]. This was followed by serology analysis using polyvalent anti-*Salmonella* somatic and flagellar sera (Probac®, Brazil). PCR amplification of the *ompC* gene was performed using the OMPCF and OMPCR primers [35] for confirmation of the genus [36]. Positive controls were provided by the Oswaldo Cruz Foundation.

## Molecular serotyping of *L. monocytogenes* and *Salmonella*

*L. monocytogenes* subtyping was performed using multiplex PCR and primers lmo0737, lmo1118, ORF2819 and ORF2110, as per methods described by Doumith *et al*. [11] for differentiation into the four main lineages.

Salmonella spp. serovars were identified via WGS at the Laboratory of Bacterial Resistance and Therapeutic Alternatives at the Institute of Biomedical Sciences of the University of São Paulo (USP–São Paulo/SP). Genomic DNA extraction was performed using the PureLink™ Genomic DNA Mini Kit (Life Technologies, CA, USA), and used for library preparation using the Nextera DNA Flex Library Prep Kit (Illumina, San Diego, CA). Subsequently, the DNA quantified by Qubit 2.0 fluorometer (Life Technologies) was sequenced using the Illumina NextSeq PE instrument (Illumina Inc., San Diego, CA) using a paired-end (75bp) library. The short reads were trimmed with TrimGalore v0.6.5 (https://github.com/FelixKrueger/TrimGalore) and *de novo* assembly was performed using Unicycler v.0.4.8 [37]. Genome annotations were performed using NCBI PGAP v.3.2 (http://www.ncbi.nlm.nih.gov/genome/annotation_prok/), and serovar was predicted via SeqSero2 analysis [38].

## Pulsed-field gel electrophoresis of *L. monocytogenes*

PFGE was performed as per guidelines recommended by the Centers for Disease Control and Prevention [39]. The restriction enzyme *AscI* (Invitrogen®, Lithuania) was used at a concentration of 10 U/μL. *Salmonella* ser. Braenderup H9812 subjected to digestion with *XbaI* (Roche®, Germany) at a concentration of 10 U/μL was used as the standard, and was provided by the Enterobacteria Laboratory of the Oswaldo Cruz Foundation. The BioNumerics 7.7. software (Applied Maths) was used to analyze the fragments, with a Dice similarity coefficient of 1.5%. The dendrogram was constructed by analyzing the unweighted pair group method with arithmetic mean (UPGMA) clusters.

## Analysis of the *inlA* gene sequence of *L. monocytogenes*

The sequencing of the *inlA* gene (2400bp) was performed following the protocol described by Poyart *et al*. 1996 [17]. Two gene fragments of 1157 bp and 760 bp were amplified, using primers O1, O2, O3, and O4 [17]. PCR products were purified using the PureLink kit (Invitrogen®, USA), following which they were quantified by using high mass marker (Invitrogen®, Lithuania) and were sequenced using the ABI3500 sequencer (Applied Biosystems).

## Cell adhesion and invasion assays for *L. monocytogenes*

The Caco-2 human colon adenocarcinoma cell line was used. The cells were provided by Prof. Elaine Cristina Pereira De Martinis, from the Faculty of Pharmaceutical Sciences of Ribeirão

Preto, USP. Cells were cultured as per methods described by Gaillard *et al.* 1987 [40]. Wells in polypropylene plates (Kasvi®) were seeded with cells at a final density of $1.0 \times 10^5$ cells per well. Adhesion assays were conducted as per protocols described by Moroni *et al.* 2006 [41]. The wells seeded with the cells were inoculated with a bacterial suspension using a volume adjusted to obtain a multiplicity of infection (MOI) of 100. After adding the Dulbecco's modified Eagle medium (DMEM) (Gibco®, USA) supplemented with 10% fetal bovine serum (Gibco®, USA), the plates were incubated at 37˚C for 2 h in a 5% $CO_2$ atmosphere. The wells were rinsed three times with 1× phosphate-buffered saline (PBS; Laborclin®, Brazil). The cells were then subjected to treatment with a lysis buffer solution containing 0.1% Triton-X100 (Sigma-Aldrich®, USA), and incubated for 10 min under the aforementioned conditions. Viable bacterial cells were counted by seeding serial dilutions on BHI agar (Difco®, France) incubated at 37˚C for 24 h. The counts were expressed as colony forming units (CFU)/mL. All tests were performed in triplicate. Adhesion of bacteria to Caco-2 cells was calculated (%) as (number of adherent cells × 100) / number of cells adhering to the well [41]. Invasion tests were performed as per methods described by Gaillard *et al.* 1987 [40] and Moroni *et al.* 2006 [41]. Cells were added to each well at an MOI of 100. Following incubation, the cells were subjected to washing steps with PBS as described above for the adhesion assay. After subjection to washing steps, 1 mg/mL gentamicin (Sigma-Aldrich®, Israel) was added to the wells and incubation was performed at 37˚C for 1 h in a 5% $CO_2$ atmosphere. The lysis buffer solution was added to each well, and viable bacterial cells were counted as described above. All tests were performed in triplicate. The percentage of bacterial invasion in Caco-2 cells was determined by using the following formula: % adhesion = (number of internalized recovered cells × 100) / number of cells adhering to the well.

## Antibiogram and assessment of antimicrobial resistance genes

The AMR of *L. monocytogenes* isolates was evaluated by performing the disk diffusion assay using Mueller Hinton agar (Acumedia®, USA), as per methods described by the Clinical and Laboratory Standards Institute [42]. The antibiotics tested were ampicillin (10 μg), ciprofloxacin (5 μg), chloramphenicol (30 μg), doxycycline (30 μg), erythromycin (15 μg), gentamicin (10 μg), sulfonamides (300 μg), and tetracycline (30 μg). The interpretation of the zone of inhibition diameters was based on the standards prescribed for *Staphylococcus* spp. [43] which were defined by CLSI M100 [42], with the exception of standards for erythromycin and ampicillin, for which the standards defined by the European Committee on Antimicrobial Susceptibility Testing [44] were used for *L. monocytogenes*. The identification of resistance genes in *L. monocytogenes* isolates was performed using PCR. Genes related to the resistance to tetracycline (*tetA*, *tetB*, *tetC*, *tetM*), macrolides (*ermA*, *ermB*, *ermC*, *ereA*), amphenicols (*cat1*, *cmlA*), sulfonamides (*sull*), beta-lactams (*ampC*, *blaSHV*), and aminoglycosides (*aac (3)-I*) were investigated. Reactions were performed in a final volume of 25 μL, according to the conditions described in the studies reported in Table 1 for each pair of primers.

The AMR of *Salmonella* spp. was evaluated by performing the disk diffusion assay using Mueller Hinton agar (Acumedia®, USA), according to the CLSI protocol [42]. The tested antibiotics were nalidixic acid (30 μg), amoxicillin (10 μg), ampicillin (10 μg), cephalothin (30 μg), cefazolin (30 μg), ceftazidime (30 μg), ciprofloxacin (5 μg), chloramphenicol (30 μg), colistin (10 μg), doxycycline (30 μg), gentamicin (10 μg), sulfonamides (300 μg), and tetracycline (30 μg). To interpret the results, the standards prescribed for *Enterobacteriaceae* defined by the CLSI M100 [42] were used. The presence of resistance genes was evaluated via WGS using the ResFinder v.4.0. The presence of genes related to the resistance to the following antimicrobials was investigated: quinolones, tetracycline, nitroimidazole, sulfonamides,

**Table 1. Primers used to investigate antimicrobial resistance genes in *Listeria monocytogenes* isolates.**

| Gene | Primer | Nucleotide sequence (5'-3') | Size (pb) | Reference |
|---|---|---|---|---|
| *aac(3)-I* | aac(3)-I-F | ACCTACTCCCAACATCAGCC | 157 | Van *et al.*, 2008 [45] |
| | aac(3)-I-R | ATATAGATCTCACTACGCGC | | |
| *ampC* | AmpC-For | TTCTATCAAMACTGGCARCC | 550 | Schwartz *et al.*, 2003 [46] |
| | AmpC-Rev | CCYTTTTATGTACCCAYGA | | |
| *bla*SHV | blaSHV-F | TCGCCTGTGTATTATCTCCC | 768 | Van *et al.*, 2008 [45] |
| | blaSHV-R | CGCAGATAAATCACCACAATG | | |
| *ermA* | ermA-F | TCTAAAAAGCATGTAAAAGAA | 645 | Sutcliffe *et al.*, 1996 [47] |
| | ermA-R | CTTCGATAGTTTATTAATATTAGT | | |
| *ermB* | ermB-F | GAAAAGGTACTCAACCAAATA | 639 | Sutcliffe *et al.*, 1996 [47] |
| | ermB-R | AGTAACGGTACTTAAATTGTTTAC | | |
| *ermC* | ermC-F | TCAAAACATAATATAGATAAA | 642 | Sutcliffe *et al.*, 1996 [47] |
| | ermC-R | GCTAATATTGTTTAAATCGTCAAT | | |
| *ereA* | ere(A)-F | GCCGGTGCTCATGAACTTGAG | 419 | Van *et al.*, 2008 [45] |
| | ere(A)-R | CGACTCTATTCGATCAGAGGC | | |
| *cat1* | CATIF | AGTTGCTCAATGTACCTATAACC | 547 | Van *et al.*, 2008 [45] |
| | CATIR | TTGTAATTCATTAAGCATTCTGCC | | |
| *cmlA* | cmlA-F | CCGCCACGGTGTTGTTGTTATC | 698 | Van *et al.*, 2008[45] |
| | cmlA-R | CACCTTGCCTGCCCATCATTAG | | |
| *sul* | sulI-F | TTCGGCATTCTGAATCTCAC | 822 | Van *et al.*, 2008 [45] |
| | sulI-R | ATGATCTAACCCTCGGTCTC | | |
| *tetA* | tet(A)-F | GTGAAACCCAACATACCCC | 887 | Van *et al.*, 2008 [45] |
| | tet(A)-R | GAAGGCAAGCAGGATGTAG | | |
| *tetB* | tet(B)-F | CCTTATCATGCCAGTCTTGC | 773 | Van *et al.*, 2008 [45] |
| | tet(B)-R | ACTGCCGTTTTTTCGCC | | |
| *tetC* | tet(C)-F | ACTTGGAGCCACTATCGAC | 880 | Van *et al.*, 2008 [45] |
| | tet(C)-R | CTACAATCCATGCCAACCC | | |
| *tetM* | tet(M)-1 | GTTAAATAGTGTTCTTGGAG | 700 | Aarestrup *et al.*, 2000 [48] |
| | tet(M)-2 | CTAAGATATGGCTCTAACAA | | |

macrolides, rifampicin, glycopeptides, colistin, trimethroprim, fusidic acid, aminoglycosides, beta-lactams, oxazolidinone, and fosfomycin. Chromosomal mutations related to AMR in the *gyrA*, *gyrB*, *pmrA*, *pmrB*, *parC*, *parE*, and *16s_rrsD* genes were investigated.

## *In vitro* biofilm formation capacity

*In vitro* biofilm formation capacity of *L. monocytogenes* and *Salmonella* spp. isolates was evaluated by performing incubation at 37°C for 24 h and incubation at 12°C for 168 h (7 days). Tests were performed using polystyrene titration microplates as per methods described by Djordjevic *et al.* 2002 [49] and modified by Borges *et al.* 2018 [50]. TSB broth (Kasvi®, Italy) was used to prepare bacterial suspensions at a final density of 3 x $10^8$ CFU/mL. All tests were performed in triplicate. After the performance of preparation, incubation, washing steps, staining procedures, and resuspension of each sample, the optical density (OD) of each well was measured using an ELx800 enzyme-linked immunosorbent assay reader (Biotek Instruments) at 490 nm. The OD value for each isolate was determined as the arithmetic mean of the absorbance readings obtained from the three wells. This value was compared with that of the negative control (ODn). The classification proposed by Stepanović *et al.* 2000 [51] was used to determine the capacity and intensity of biofilm formation. Statistical analysis of data for

comparison of the capacity of biofilm formation in the two conditions tested was performed using the SAS software (SAS Inc.).Normality was analyzed using the Shapiro-Wilk test followed by the paired t-test for means comparison between the two groups.

## Results and discussion

### Detection of microorganisms in poultry and cattle slaughterhouses

**Detection of *L. monocytogenes*.** A total of 14 *L. monocytogenes* isolates were detected from 287 swabs collected from poultry and cattle slaughterhouses (4.87%). All isolates were detected in poultry slaughterhouses (A, B, and C), which represented 169 swabs (8.28%). *L. monocytogenes* was not detected in any of the 118 samples collected from the cattle slaughterhouses (D and E). The detection points for the isolates are listed in Table 2. In the Federal District and State of Goiás region, reports have described the presence of *L. monocytogenes* isolated from different sources [52, 53]. These findings support the results of the present study. The absence of *L. monocytogenes* in cattle slaughterhouses differs from the findings reported by Palma *et al.* 2016 [53], who described the presence of *L. monocytogenes* in samples obtained from cattle slaughterhouses. The difference between the studies may be related to the small number of industries available and the sampling visits that were conducted, which was due to the hesitation of the facilities to participate in the study. The difference may also reflect the adequate hygiene and sanitation conditions prevalent in these establishments.

The detection of *L. monocytogenes* in poultry slaughterhouses is similar to the findings reported in previous studies described in Brazil [4, 54–56]. The detection of *L. monocytogenes* at the sampling points described in the present study agrees with previously reported detection on stainless steel tables [4, 55], mats [4], chutes [56], and drains [12, 28]. The usual definition of persistence in the context of foodborne pathogens is the repeated isolation of strains that are identical subtypes identified on different days [2]. In this study, strains were considered persistent when detected at the same sampling location after at least one sanitation process. All collections were performed after the environment was subjected to sanitation procedures at least once, suggesting the persistence of the microorganism, possibly due to the ability to adhere to surfaces and to form biofilms [57–59]. At one collection point (a chute in slaughterhouse C), *L. monocytogenes* was detected in two samples collected on different days, suggesting the presence of biofilms at the site. In two different studies conducted in the United States, Berrang *et al.* [28, 60] performed serial collection from the same points within the slaughter plant at different times to characterize the presence of persistent *L. monocytogenes* isolates in the environments of poultry slaughterhouses. The authors concluded that the persistence of this microorganism in drains was related to the formation of biofilms [60]. Similar to the methods described in the present study, Sereno *et al.* 2019 [61] performed serial swab collections of walls, drains, tables, mats, and floors in a swine slaughterhouse in the southern region of Brazil. The persistence of *L. monocytogenes* strains was detected at the same sampling point over time and was related to the presence of biofilms at said location. Areas of bacterial persistence include drains [28, 60] and a mat [61]. The present study also detected the presence of *L. monocytogenes* at these sites, although there was no detection at the same site more than once, except for the chute.

### Detection of *Salmonella* spp.

One *Salmonella* sp. was detected in 287 (0.34%) swabs collected from the cattle and poultry slaughterhouses. The isolate originated from a drain sample that was collected from a dirty area of poultry slaughterhouse B. This is the first investigation of *Salmonella* spp. in a processing plant of a poultry slaughterhouse located in the Federal District. However, other reports

**Table 2. Detection of *Listeria monocytogenes* in swabs collected from poultry slaughterhouse facilities, equipment and utensils located in the Federal District and State of Goiás, Brazil.**

| Region and establishment | DF | DF | DF | DF | DF | GO | GO | DF | | | | |
|---|---|---|---|---|---|---|---|---|---|---|---|---|
| Identification | A | A | A | A | A | C | C | B | | | | |
| Swab collection points | Collection 1 | Collection 2 | Collection 3 | Collection 4 | Collection 5 | Collection 6 | Collection 7 | Collection 8 | Total *Listeria monocytogenes* strains | Total no. of swabs obtained from each collection point | Percentage of positive samples at each collection point (%) | Identification of isolates detected at each collection point |
| | No. of swabs = 16 | No. of swabs = 16 | No. of swabs = 16 | No. of swabs = 15 | No. of swabs = 25 | No. of swabs = 25 | No. of swabs = 25 | No. of swabs = 31 | | | | |
| **Facilities** | | | | | | | | | **4** | **56** | 7.14 | |
| Floor in clean area | 0 | 0 | 0 | 0 | 0 | 0 | 0 | 0 | 0 | 1 | 0.0 | |
| Drains in clean area | 0 | 0 | 1 | 0 | 0 | 0 | 1 | 1 | 3 | 38 | 7.89 | 63A-1;69A-2; 117A-3 |
| Drains in dirty area | 0 | 0 | 0 | 0 | 0 | 0 | 0 | 0 | 0 | 2 | 0.0 | |
| Walls in clean area | 0 | 0 | 0 | 0 | 0 | 0 | 1 | 0 | 1 | 13 | 0.0 | 76A-2 |
| Walls in dirty area | 0 | 0 | 0 | 0 | 0 | 0 | 0 | 0 | 0 | 2 | 0.0 | |
| **Equipment and utensils** | | | | | | | | | **10** | **113** | 8.84 | |
| Evisceration table | 0 | 0 | 0 | 0 | 0 | 0 | 2 | 0 | 2 | 35 | 5.71 | 74A-2;77A-2 |
| Mats in clean area | 0 | 0 | 0 | 0 | 0 | 3 | 0 | 0 | 3 | 33 | 9,09 | 45A-2; 52A-2; 59A-2 |
| Chutes of meat | 0 | 0 | 0 | 0 | 0 | 2 | 3 | 0 | 5 | 13 | 38.46 | 42A-2; 54A-2; 72A-2; 78A-2; 88A-2 |
| Chutes of carcass | 0 | 0 | 0 | 0 | 0 | 0 | 0 | 0 | 0 | 14 | 0.0 | |
| Chutes of bones and viscera | 0 | 0 | 0 | 0 | 0 | 0 | 0 | 0 | 0 | 5 | 0.0 | |
| Hooks | 0 | 0 | 0 | 0 | 0 | 0 | 0 | 0 | 0 | 6 | 0.0 | |
| Machinery | 0 | 0 | 0 | 0 | 0 | 0 | 0 | 0 | 0 | 7 | 0.0 | |
| **Total no. of *L. monocytogenes* isolates obtained in each collection** | 0 | 0 | 1 | 0 | 0 | 5 | 7 | 1 | **14** | | | |
| **Total no. of swabs obtained in each collection** | 16 | 16 | 16 | 15 | 25 | 25 | 25 | 31 | | 169 | | |

have described the occurrence of *Salmonella* spp. in poultry carcasses/viscera produced and/or sold in this region [36]. The low detection in poultry slaughterhouses and absence in cattle slaughterhouses may reflect the efficiency of self-control, disinfection, and sanitation programs in these industries, the reduced number of visits resulting from the hesitation of industries to allow internal access for sample collection, and the implementation of control programs by the government with the aim of reducing the incidence of *Salmonella* spp. to protect consumers from considerable risks [62–64].

### Serotyping analysis by PCR and evaluation of genetic variability of *Listeria monocytogenes* by PFGE

All 14 isolates were classified via PCR as 1/2a or 3a serotypes, belonging to lineage II [65–67], with amplification of the *lmo0737* gene (691pb). These results corroborate reports available in the literature that highlight serotype 1/2a, along with serotypes 1/2b and 1/2c, as one of the most commonly detected serotypes in food samples and food production environments of animal origin [68]. The polymorphic results obtained via PFGE identified 11 distinct pulsotypes that were grouped into four clusters (I, II, III and IV) with a similarity threshold of 80% [69] (Figs 1 and S1). The sample collection points of origin from each pulsotype are shown in Fig 2. To analyze strain dissemination within industries and in the studied regions, clonal variations were considered only for those isolates that presented 100% identity with each other [70]. Pulsotype 9 was the most frequently detected (3/14 isolates), followed by pulsotype 5 (2/14 isolates). The same pulsotype was detected at different sampling points within the same industry (pulsotype 5, detected in a meat chute (72A-2) and an evisceration table (77A-2); and pulsotype 9, detected on a mat in the clean area (59A-2) and a meat chute (78A-2), both present on slaughterhouse C located in Goiás State; Table 2), suggesting the dissemination of these two clones within this establishment. Pulsotype 9 was detected at different sites and at different visits within this slaughterhouse. Though sanitization was performed between sample collections, the fact that the same strain was detected in different collections within the same establishment indicates the presence of biofilms. This finding corroborates those reported by Berrang *et al.* 2005 [28], Berrang *et al.* 2010 [60], Camargo *et al.* 2015 [13], and Sereno *et al.* 2019 [61]. Pulsotype 9 was also detected in a drain sample obtained from slaughterhouse B, located in the Federal District, suggesting regional dissemination of the strain. These results highlight the importance of using molecular typing methods with high discriminatory power, such as PFGE, to investigate the dissemination and persistence of strains within food processing plants

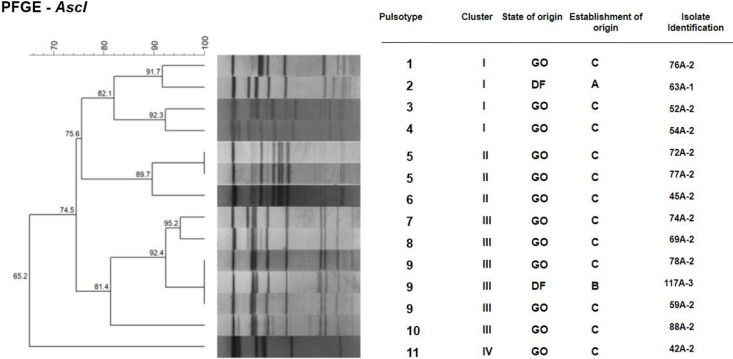

**Fig 1. Dendrogram and PFGE patterns of 14 isolates of *Listeria monocytogenes* restricted with *AscI*.** The data were analyzed using the BioNumerics software. Discrimination of pulsotypes, clusters, origins of isolates (Federal District, DF; or Goiás, GO), establishment of origin (A, B or C), and isolate identification are presented.

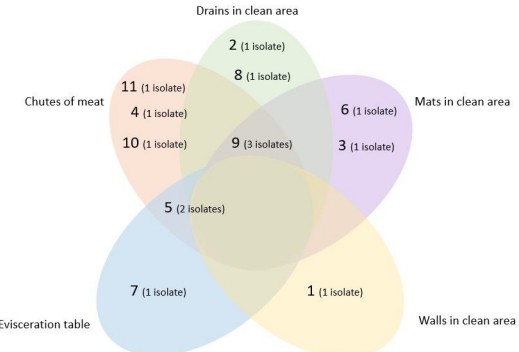

**Fig 2.** Sample collection points of the 11 *L. monocytogenes* pulsotypes detected in poultry slaughterhouses A, B and C located in Federal District and Goiás. The numbers in parenthesis represent the number of isolates belonging to each pulsotype, totaling 14 isolates.

[12, 13, 71, 72]. These typing methods are valuable when WGS analysis cannot be performed. Similar to the results obtained in the present study in relation to the use of PFGE, Camargo *et al.* 2015 [13] described the serial collection of samples during different visits from environmental locations, cuts of meat, and the hands of employees working in a cattle slaughterhouse in the state of Minas Gerais. This aided the identification of cross-contamination between these points and helped determine the persistence of specific pulsotypes. The results of the present study confirm the persistence of two pulsotypes (5 and 9) in slaughterhouse C and possible regional strain dissemination (pulsotype 9). However, due to the low number of isolates identified in slaughterhouses A and B, it was not possible to confirm the persistence of isolates within these establishments. However, the detection of isolates before the commencement of operations in the slaughterhouses (pre-sanitization) and after slaughter (post-sanitization) suggests the continuous presence of this microorganism in existing biofilms in these facilities.

## *Salmonella* sp. serotyping by WGS analysis

WGS analysis of *Salmonella* sp. isolate identified it as *Salmonella enterica* serovar Minnesota belonging to ST548 [73]. This finding differs from those reported in other studies in the investigated regions, which indicated a higher occurrence of serovar Enteritidis [36, 74, 75]. *S.* Minnesota ST548 derived from chicken carcasses, chicken feet, and mechanically recovered meat have been reported in Brazil (São Paulo and Minas Gerais, São Paulo, and Federal District, respectively) [76]. On the other hand, this is the first report of its presence in a poultry slaughterhouse located in the Federal District.

## Sequencing analysis of the *inlA* gene in *L. monocytogenes* isolates

PMSCs were not detected in any of the 14 *L. monocytogenes* isolates investigated. These results are similar to the results reported by Medeiros *et al.* 2020 [77]. The authors reported the absence of PMSCs in *L. monocytogenes* isolates in samples collected from poultry slaughterhouse drains in the Federal District. However, results of the present study differ from those reported previously by Manuel *et al.* 2015 [78] in the United States and by Camargo *et al.* 2019 [79] in Brazil. Both studies reported a higher occurrence of PMSCs in strains isolated from food samples and food production environments. The occurrence of PMSCs in *L. monocytogenes* isolates is associated with the occurrence of attenuated invasion phenotypes [18, 80]. Isolates from human listeriosis have less frequent PMSCs in the *inlA* gene than isolates obtained from food [78, 79, 81]. The present report indicating the absence of PMSCs is an important

finding for public health, since the integrity of the *inlA* gene is necessary to promote the internalization of this pathogen in host cells [18, 82].

## Cell invasion and adhesion assays

All 14 *L. monocytogenes* isolates demonstrated adherence to the Caco-2 cell surface, with an adhesion capacity that varied from 17.38% to 57.14%. Seven of the fourteen isolates (50%) showed ability to invade Caco-2 cells, with an invasion capacity varying from 5.16% to 34.27% (Fig 3). The isolates that exhibited invasion ability belonged to *clusters* I and III, and pulsotypes 1 (76A-2), 4 (54A-2), 7 (74A-2), 9 (59A-2, 78A-2 and 117A-3), and 10 (88A-2) (Fig 3). The three isolates belonging to pulsotype 9 could invade Caco-2 cells, although presenting with different capacities. The invasion capacity observed in seven of the fourteen isolates, in which the presence of PMSCs was not detected, corroborates with the findings reported by Nightingale *et al.* 2005 [18]. The latter authors reported that isolates without PMSCs could invade Caco-2 cells with an ability that was significantly more than isolates with PMSCs in the *inlA* gene. However, the results differed from those reported for the other seven isolates. Despite the absence of PMSCs, these isolates could not invade Caco-2 cells. Several studies have demonstrated the central role of *inlA* in cell invasion [18, 81, 83]. However, other elements such as internalin B [84] and listeriolysin O [85] may also be important for the internalization of *L. monocytogenes* in different cell types. Furthermore, differences in invasion capacity among isolates with the complete gene *inlA* have been described, indicating that other factors may be related to the efficiency of invasion of this pathogen [86].

## Antibiograms and detection of resistance genes

**L. monocytogenes.** Resistance or intermediate sensitivity was detected for seven of the eight antimicrobials tested (Table 3). Six of the fourteen isolates (42.9%) were sensitive to all the tested antibiotics. Eight isolates (57.1%) displayed resistance or intermediate sensitivity. The absence of resistance to ampicillin and chloramphenicol corroborates the results observed by other authors in isolates obtained from food sources and food production environments [53, 87–89]. Surveillance of the occurrence of ampicillin resistance is important for public health, since this drug is a treatment of choice for human listeriosis [90]. The sensitivity to chloramphenicol can be explained by the ban on the use of this antibiotic for the production of food of animal origin in Brazil since 2003 [91]. The results of the present study on the resistance and intermediate sensitivity towards ciprofloxacin, erythromycin, and gentamicin (42.8%, 35.7%, and 35.7% of *L. monocytogenes* isolates, respectively) contrasted with those observed by other authors, such as Teixeira *et al.* 2020 [89], who reported the absence of resistance to the three

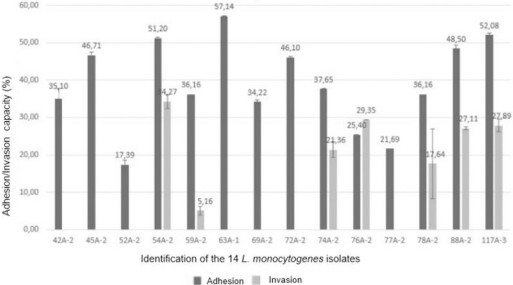

**Fig 3. Results of invasion and cell adhesion tests using Caco-2 cells for 14 *Listeria monocytogenes* isolates.** The values shown represent the average of the results, and the bars represent the standard deviation.

**Table 3. Antibiogram constructed for 14 *Listeria monocytogenes* isolates obtained from poultry slaughterhouses located in the region of the Federal District and State of Goiás, based on the results obtained via disk diffusion assay for antimicrobial resistance (CLSI, 2020).**

| Antimicrobial and class | No. of resistant isolates (%) | No. of isolates with an intermediate sensitivity (%) | No. of sensitive isolates (%) | Total no. of intermediate resistance-displaying isolates (%) |
|---|---|---|---|---|
| Sulfonamides (sulfonamide) | 8 (57.1) | 0 (0.0) | 6 (42.9) | 8 (57.1) |
| Erythromycin (macrolide) | 4 (28.6) | 1 (7.1) | 9 (64.3) | 5 (35.7) |
| Ciprofloxacin (quinolone) | 3 (21.4) | 3 (21.4) | 8 (57.2) | 6 (42.9) |
| Gentamicin (aminoglycoside) | 4 (28.6) | 1 (7.1) | 9 (64.3) | 5 (35.7) |
| Chloramphenicol (chloramphenicol) | 0 (0.0) | 4 (28.6) | 10 (71.4) | 4 (28.6) |
| Tetracycline (tetracycline) | 1 (7.1) | 1 (7.1) | 12 (85.8) | 2 (14.3) |
| Doxycycline (tetracycline) | 1 (7.1) | 0 (0.0) | 13 (92.9) | 1 (7.1) |
| Ampicillin (beta-lactamases) | 0 (0.0) | 0 (0.0) | 14 (100) | 0 (0.0) |

antibiotics in isolates obtained from beef cuts in the state of Mato Grosso, Brazil. The findings of the present study on resistance to gentamicin and erythromycin indicate a public health concern, since these are the drugs of choice for the treatment of listeriosis in specific cases, such as the use of gentamicin associated with penicillin as an alternative to ampicillin, and the use of erythromycin in the treatment of infections in pregnant women [90, 92]. The low occurrence of resistance to doxycycline detected in this study corroborates the findings reported by Vitas *et al.* 2007 [93], who reported a low occurrence of resistance in isolates of *L. monocytogenes* obtained from food and human clinical cases in Spain. The low detection of resistance to tetracycline (7.1% of the isolates) is in agreement with the findings reported by Haubert *et al.* 2016 [88] for isolates obtained from food and food production environments in the southern region of Brazil. However, the findings of the present study differ from those described by Palma *et al.* 2016 [53] and Camargo *et al.* 2015 [87], in which resistance to tetracycline was not detected in isolates obtained from samples of beef cuts, cattle slaughterhouse environments, and clinical cases in Brazil. The highest occurrence of AMR detected in this study was in relation to sulfonamides, a characteristic which was found in eight isolates (57.1%). These findings corroborate the results reported by other authors, who also reported a pronounced prevalence of resistance to this class of antimicrobials in isolates obtained from food and food production environments [53, 89, 94, 95]. Sulfonamides associated with trimethoprim are considered as the second-choice of treatment for human listeriosis [90]. Therefore, the detection of resistance to this class of antimicrobials may indicate a risk to human health.

The results of the detection of AMR genes are shown in Table 4. The *ermA*, *ermC*, *cat1*, *sulI*, *aaC (3)-1*, *cmlA*, *ereA*, *SHV*, and *ampC* genes were not detected.

The detection of the *ermB* gene was in accordance with the finding reported by Haubert *et al.* [88], who reported the presence of this gene in an isolate of *L. monocytogenes* obtained from a poultry slaughterhouse environment in the southern region of Brazil. Only two of the four isolates of *L. monocytogenes* that presented with an erythromycin resistance phenotype in the disk diffusion assay (Table 3) harbored one of the erythromycin-related resistance genes investigated. It could be possible that the isolates that did not harbor *ermA*, *ermB* or *ermC* genes

**Table 4. Results of antibiogram susceptibility test and antimicrobial resistance genes for the 14 *L. monocytogenes* isolates.**

| *L. monocytogenes* isolate identification | Pulsotype | Region and establishment identification | Swab collection point in the industry | Antibiogram results of resistance or intermediate sensibility | Antimicrobial resistance genes |
|---|---|---|---|---|---|
| 42A-2 | 11 | GO/C | Chutes of meat | Sensible to all tested bases | No gene targeted in the study detected |
| 45A-2 | 6 | GO/C | Mats in clean area | CIP* ERI GEN* SUL VAN* | *tet(M) tet(C)* |
| 52A-2 | 3 | GO/C | Mats in clean area | CIP* CLO* ERI GEN SUL TET* VAN | *ermB tet(C)* |
| 54A-2 | 4 | GO/C | Chutes of meat | CIP CLO* DOX ERI GEN SUL TET VAN* | *ermB tet(M) tet(C)* |
| 59A-2 | 9 | GO/C | Mats in clean area | Sensible to all tested bases | *tet(B) tet(C)* |
| 63A-1 | 2 | DF/A | Drains in clean area | Sensible to all tested bases | *tet(C)* |
| 69A-2 | 8 | GO/C | Drains in clean area | Sensible to all tested bases | *tet(B)* |
| 72A-2 | 5 | GO/C | Chutes of meat | CIP CLO* ERI* GEN SUL VAN | *tet(C)* |
| 74A-2 | 7 | GO/C | Evisceration table | SUL | *tet(B)* |
| 76A-2 | 1 | GO/C | Walls in clean area | Sensible to all tested bases | *tet(C)* |
| 77A-2 | 5 | GO/C | Evisceration table | SUL | *tet(C)* |
| 78A-2 | 9 | GO/C | Chutes of meat | Sensible to all tested bases | *tet(C)* |
| 88A-2 | 10 | GO/C | Chutes of meat | CIP* CLO* ERI GEN SUL VAN* | *tet(B) tet(C)* |
| 117A-3 | 9 | DF/B | Drains in clean area | CIP SUL TEC* | *tet(B) tet(C)* |

* Antimicrobial agents with intermediate sensitivity.

might have harbored genes related to the development of other mechanisms of resistance to this antimicrobial, such as those pertaining to the presence of *msr(A)* or *mef(A)* genes [8]. Alternatively, the resistance of these isolates may be attributable to chromosomal mutations [95].

The presence of the *tetM* gene in two isolates (14.28%) was similar to the findings reported by Bertrand *et al.* [96], who reported the presence of this gene in isolates exhibiting phenotypic resistance to tetracycline in strains obtained from human clinical samples and swine and poultry slaughterhouse environments from Belgium and France. Several studies have highlighted the *tetM* gene as the genotype that is most commonly associated with the development of tetracycline resistance in *Listeria* isolates [96, 97]. These findings differ from findings of the present study that suggested that *tetC* was the most prevalent gene related to tetracycline resistance. Despite the detection of *tetC* in eleven of the fourteen isolates (78.6%), only two isolates displayed resistance or intermediate sensitivity to tetracycline in the disk diffusion assay (Table 3). One of these two isolates harbored two of the four genes related to the development of resistance to this drug (*tetC* and *tetM*), and was classified as a resistant isolate based on the

antibiogram. The other isolate harbored only the *tetC* gene and showed intermediate sensitivity to this antimicrobial. The other nine isolates with the *tetC* gene did not demonstrate resistance to tetracycline in the disk diffusion assay. Therefore, the unique presence of *tetC* in the isolates did not confer a resistance phenotype to tetracycline, suggesting that the genes did not necessarily trigger the resistance phenotype, which might be related to the presence of mutations that could lead to gene dysfunction [98].

Despite the occurrence of AMR to gentamicin in five of the fourteen isolates in the disk diffusion assay (Table 3), the presence of the *aac(3)-1* gene was not detected in any isolate. It could be possible that resistance to gentamicin in these isolates was related to the presence of other genes not investigated in this study, since more than 170 genes related to resistance to aminoglycosides have been described in bacteria [8]. In the present study, resistance to sulfonamides was observed in eight of fourteen isolates in the disk diffusion assay, but no isolates harbored the *sulI* gene. It could be possible that this resistance was associated with other mechanisms, such as the presence of other genes related to sulfonamide resistance, namely *sul2*, *folP*, or *thyA* genes, with the description of the last two genes reported in *L. monocytogenes* [99].

Only intermediate sensitivity to chloramphenicol was detected in four of fourteen isolates in the disk diffusion assay. The *cmlA* and *cat1* genes were not detected. The intermediate sensitivity observed in this study could be related to cross-resistance due to the presence of other genes that were not investigated, such as *floR* [100]. Notably, in Brazil, this antimicrobial has been banned from use in animal production since 2003 [91]. Selection pressure was not a factor. The absence of the *ampC* and *bla*$_{SHV}$ genes is consistent with the absence of the ampicillin resistance phenotype in the disk diffusion assay.

Four of the fourteen isolates (28.5%) showed resistance to three or more classes of antimicrobials and were classified as multidrug-resistant isolates [101]. All were isolated from establishment C. Two isolates (54A-2 and 88A-2) showed adherence and invasion in Caco-2 cells. Reports of the presence of *L. monocytogenes* multidrug-resistant isolates have increased in the literature [88, 97, 102]. All multidrug-resistant isolates detected in the present study showed resistance to antimicrobials used in the treatment of human listeriosis (gentamicin and erythromycin) [90]. There is a possibility of the transfer of genes from multidrug-resistant isolates to others. Further studies

**Table 5. Antibiogram and detection of antimicrobial resistance genes from the *Salmonella enterica* serovar Minnesota isolate from a poultry slaughterhouse located in the Federal District, carried out using the disk diffusion assay (CLSI, 2020) and WGS, respectively.**

| Antibiogram (tested drugs) | Antibiogram result | Antimicrobial resistance genes |
|---|---|---|
| Nalidixic acid | R | *qnrB19* |
| Amoxicillin | R | *bla*$_{CMY-2}$ |
| Ampicillin | R | *bla*$_{CMY-2}$ |
| Cephalothin | R | *bla*$_{CMY-2}$ |
| Cefazoline | R | *bla*$_{CMY-2}$ |
| Ceftazidime | R | *bla*$_{CMY-2}$ |
| Ciprofloxacin | S | - |
| Chloramphenicol | S | - |
| Colistin | S | - |
| Doxycycline | I | - |
| Gentamicin | S | *aac(6')-1aa* |
| Sulfonamides | R | *sul2* |
| Tetracycline | R | *tetA* |

S, sensitive; I, intermediate; R, resistant.

are warranted to elucidate the resistance mechanisms observed in these isolates and their potential as a source of transfer of resistance to other microorganisms.

***Salmonella enterica* Minnesota.**   Resistance or intermediate sensitivity of *Salmonella* Minnesota was detected in nine of thirteen antimicrobials tested (Table 5). The resistance to tetracycline and ampicillin, and the sensitivity to ciprofloxacin and gentamicin were similar to the results obtained by Dantas *et al.* [103]. The authors reported the same pattern of resistance in isolates of *Salmonella* spp. obtained from a poultry slaughterhouse environment in the state of São Paulo. The detection of resistance to ampicillin, cephalothin, and sulfonamide bases, in addition to sensitivity to ciprofloxacin, is comparable to the findings reported by Cunha-Neto *et al.* 2018 [104] who investigated isolates of *Salmonella* spp. derived from chicken carcasses in a slaughterhouse in the state of Mato Grosso. However, these authors reported the detection of resistance to gentamicin and chloramphenicol, and sensitivity to nalidixic acid, which differed from the results of the present study. The detection of resistance to nalidixic acid in *Salmonella* spp. isolates has been widely reported [105–107]. Meta-analyses of studies published over a period of 20 years in Brazil have revealed the increased occurrence of resistance to quinolone in isolates derived from human samples and chicken meat [108]. Resistance to nalidixic acid may be attributable to the prolonged and widespread use of this antimicrobial in human and veterinary medicine, since this was the first drug in the quinolone class to have clinical use [109]. Similar to nalidixic acid resistance, resistance to tetracycline, sulfonamides, and beta-lactams may be related to the overuse of these antimicrobials in the different stages of the chicken meat production chain in Brazil [110].

The present study detected resistance to three generations of cephalosporins. Cephalosporins, especially those of the third and fourth generations, are important for human and animal health [111]. Resistance to cephalosporins (including third- and fourth-generation drugs) has been reported in recent years [112, 113]. The detection of resistance, especially to ceftazidime, is a public health concern and highlights the need for vigilance and the reduction of the use of antimicrobials in veterinary medicine. In recent years, the Brazilian government has restricted the use of antimicrobials in the production of animal foods, such as via implementation of a ban on the use of chloramphenicol and nitrofurans [91], colistin sulfate [114], and more recently, via the enforcement of a ban on the use of tylosin, lincomycin, and tiamulin [115].

WGS analysis of *Salmonella* isolate (GenBank accession number JABBEB000000000.1) led to the detection of the *tetA*, *sul2*, *aac (6')-Iaa*, *bla*$_{CMY-2}$, and *qnrB19* genes, as well as aided identification of a mutation in the *parC* gene (T57S, ACC →AGC, causing the mutation T→S). All detected resistance genes have been described in *Salmonella* spp. isolates in Brazil [76, 116]. The concomitant presence of plasmid-mediated quinolone resistance (PMQR) via the *qnrB19* gene, and mutations in the region determining quinolone resistance (QRDR) in the *parC* gene of topoisomerase IV was also observed. These genetic characteristics were confirmed by ascertaining resistance to nalidixic acid, but resistance to ciprofloxacin was not detected in the antibiograms. Despite this, the presence of the *qnrB19* gene and other genes related to PMQR may be responsible for the reduced susceptibility to quinolones, which may facilitate the selection of less susceptible isolates and may lead to consequent failure in the treatment with this class of drugs [117]. The detection of PMQR in this study highlights the need for surveillance of the persistence of quinolone resistance genes in the poultry production chain. The *bla*$_{CMY-2}$ gene is a plasmid gene related to the production of beta-lactamases (*pAmpC*), and is a gene that is most commonly detected globally [118]. The presence of this gene corroborates the results of resistance to beta-lactams observed in the disk diffusion assay (ampicillin, cephalothin, cefazolin, and ceftazidime). Its presence is another public health concern considering the importance of beta-lactams in human and animal health, especially third-generation cephalosporins [111]. The presence of the *aac-(6')-Iaa* gene related to the

development of resistance to aminoglycosides did not confer resistance to gentamicin, the only drug from this class of antimicrobials evaluated in the disk diffusion assay. This discrepancy between phenotype and genotype was also observed in *Salmonella* spp. isolates obtained from different samples in the poultry production chain in the states of São Paulo, Bahia, and Minas Gerais [76]. In contrast, the occurrence of resistance to tetracycline and sulfonamides is related to the presence of the *tetA* and *sul2* genes, respectively. The absence of colistin resistance genes corroborates findings reported by Monte *et al.* [76], who reported the absence of these genes in *Salmonella* spp. isolates obtained from samples of swine and poultry production chains from different regions of Brazil, including the Federal District.

The resistance profile observed in this *Salmonella* sp. isolate enabled its classification as a multidrug-resistant isolate [101]. This characteristic, associated with the presence of mobile genetic elements related to resistance development, such as the presence of PMQR and *pAmpC*, suggests a potential public health risk. Additionally, the occurrence of resistance to quinolones and cephalosporins, associated with the presence of genetic elements related to this phenotype, is of special concern, considering the importance of these drugs for human health [111].

## *In vitro* biofilm formation capacity

**L. monocytogenes.** During incubation at 37˚C for 24 h, 11 of 14 isolates (78.57%) were capable of biofilm formation and were classified as weak biofilm formers. During incubation at 12˚C for 168 h, nine of 14 isolates (64.3%) were capable of biofilm formation. All nine were deemed weak biofilm formers (Fig 4). The biofilm formation capacity of *L. monocytogenes* isolates corroborated findings reported in other studies indicating their ability to adhere to abiotic surfaces and to form biofilms on such surfaces [119–121]. The poor biofilm formation capacity of most isolates (78.57%) at 37˚C corroborated the findings reported by Harvey *et al.* 2007 [119]. The authors reported that 90% of the tested *L. monocytogenes* isolates were weak biofilm-forming bacteria. There was no significant difference (p = 0.2450) in the biofilm formation capacity of the isolates when incubated at 37˚C for 24 h and at 12˚C for 168 h. The finding that nine of the 14 isolates can form biofilms during incubation at 12˚C highlights the risks that these isolates can pose to cattle and poultry slaughter facilities in the event of failures of the hygiene and sanitation practices adopted for the environments, since this is a common temperature documented inside the industry.

There was no association between cluster/pulsotype and *in vitro* biofilm formation capacity. Pulsotypes 9 and 5, which were the most commonly detected, did not present better *in vitro* biofilm formation capacity than the other pulsotypes (Table 6). These results are similar to the

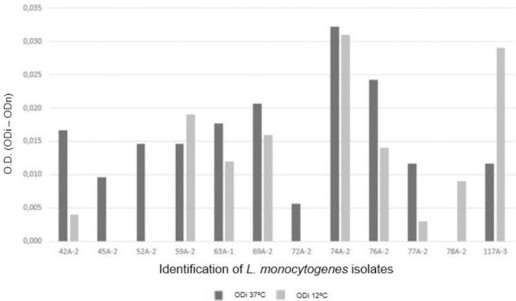

**Fig 4. Results of the *in vitro* biofilm formation capacity test performed using polystyrene microplates (Djordjevic *et al.*, 2002) and 12 *Listeria monocytogenes* isolates exhibiting biofilm formation capacity at 37˚C and/or 12˚C.** The bars represent the average value of the optical density of each test for each isolate (ODi), all performed in triplicate, subtracted from the average of the optical density of the negative control for each repetition (ODn).

findings of other studies that reported no difference in the *in vitro* ability to form biofilms between persistent and non-persistent strains [49, 119], although other studies report persistent strains as better *in vitro* biofilm formers [57, 58]. The association between adherence capacity and strain persistence in the industry remains unclear.

Three of the four multidrug-resistant isolates (52A-2, 72A-2 and 88A-2) were capable of *in vitro* biofilm formation at 37˚C, and none of them formed biofilm at 12˚C. In contrast, 13 out of the 14 isolates that harbored at least one of the resistance genes targeted in this study were capable of biofilm formation in the *in vitro* assay, at least, at one of the tested temperatures (Table 6). These results highlight the possibility of adherence of these isolates to the surfaces of these industries, posing the risk of acting as resistance gene reservoirs for other strains and species, ultimately posing a threat to public health and causing dissemination of AMR in various strains.

Most isolates investigated in the present study demonstrated the ability to form biofilms under at least one of the conditions tested. These findings, along with the findings of repeated detection of clonal variations within the same industry at different points and different visits, in addition to the identification of the same pulsotype in another slaughterhouse in another region, support the possible existence of *L. monocytogenes* biofilms in poultry slaughterhouses

**Table 6. Results of antimicrobial resistance, antimicrobial resistance genes and *in vitro* biofilm formation capacity of the 14 *L. monocytogenes* isolates.**

| *L. monocytogenes* isolate identification | Pulsotype | Antimicrobial resistance | Antimicrobial resistance genes | *In vitro* biofilm formation capacity | Classification at 37˚C | Classification at 12˚C |
|---|---|---|---|---|---|---|
| 42A-2 | 11 | Sensible to all tested bases | No gene targeted in the study detected | Yes | Weak | Weak |
| 45A-2 | 6 | CIP* ERI GEN* SUL VAN* | *tet(M) tet(C)* | Yes | Weak | Non-forming |
| 52A-2 | 3 | CIP* CLO* ERI GEN SUL TET* VAN | *ermB tet(C)* | Yes | Weak | Non-forming |
| 54A-2 | 4 | CIP CLO* DOX ERI GEN SUL TET VAN* | *ermB tet(M) tet (C)* | No | Non-forming | Non-forming |
| 59A-2 | 9 | Sensible to all tested bases | *tet(B) tet(C)* | Yes | Weak | Weak |
| 63A-1 | 2 | Sensible to all tested bases | *tet(C)* | Yes | Weak | Weak |
| 69A-2 | 8 | Sensible to all tested bases | *tet(B)* | Yes | Weak | Weak |
| 72A-2 | 5 | CIP CLO* ERI* GEN SUL VAN | *tet(C)* | Yes | Weak | Non-forming |
| 74A-2 | 7 | SUL | *tet(B)* | Yes | Weak | Weak |
| 76A-2 | 1 | Sensible to all tested bases | *tet(C)* | Yes | Weak | Weak |
| 77A-2 | 5 | SUL | *tet(C)* | Yes | Weak | Weak |
| 78A-2 | 9 | Sensible to all tested bases | *tet(C)* | Yes | Non-forming | Weak |
| 88A-2 | 10 | CIP* CLO* ERI GEN SUL VAN* | *tet(B) tet(C)* | Yes | Weak | Non-forming |
| 117A-3 | 9 | CIP SUL TEC* | *tet(B) tet(C)* | Yes | Weak | Weak |

* Antimicrobial agents with intermediate sensitivity.

located in the Federal District and State of Goiás. The detection of the biofilm formation capacity of the isolates, especially the ability documented at 12˚C and observed inside slaughterhouses, indicates the possibility that these isolates can adhere to surfaces, including those of equipment and utensils, which suggests the possibility of biofilm formation in the event of failures in appropriate implementation of hygiene procedures. This creates a potential public health risk because of possible food-associated cross-contamination.

## *Salmonella* spp.

The *Salmonella* sp. isolate was capable of biofilm formation, albeit weakly, under both conditions tested. These results corroborate those reported by Yin *et al.* 2018 [122]. The authors reported the ability of biofilm formation of *Salmonella* spp. isolates obtained from beef production environments in China at different temperatures, including 12˚C and 37˚C. The present results suggest the potential risk to public health by the *Salmonella* sp. isolate in the slaughterhouse environment, since, in addition to its multidrug-resistance characteristic, this isolate may adhere to abiotic surfaces, and may be considered a potential contaminant of processed foods.

## Conclusions

The presence of *L. monocytogenes* and *Salmonella* spp. was identified for the first time in poultry slaughterhouses located in the Federal District and in the State of Goiás, Brazil. The bacterial species were detected on equipment and utensils, especially chutes, which were the main routes of contamination by *L. monocytogenes* within the evaluated plants. Species could not be detected in cattle slaughterhouses. The consistent and repeated collection at the sampling points enabled the characterization of biofilms in the environments of the industries, which was confirmed in 12 of the 14 *L. monocytogenes* isolates and in *Salmonella* sp. via the *in vitro* biofilm formation capacity test that was performed at 12˚C and/or at 37˚C. The PFGE patterns aided the evaluation of the dissemination of specific pulsotypes both within an industry and in the studied regions. Sequencing of the *inlA* gene derived from *L. monocytogenes* isolates demonstrated the absence of PMSCs in all isolates. These results, along with the adhesion and cell invasion assay results, suggest their virulence potential. The detection of resistance to antimicrobials of public health importance in *L. monocytogenes* and *Salmonella* sp. isolates, in addition to the detection of multidrug-resistant isolates, also underlines the potential risk for the consumer due to the possibility of food-associated cross-contamination via contact with biofilms formed on abiotic surfaces. The results of this study indicate the importance of conducting research that supports surveillance in slaughterhouse environments for possible adjustments in environmental hygiene and sanitation procedures. Additionally, the detection of isolates with pathogenic potential related to the capability of cell invasion, the ability to form biofilms, and AMR development indicates the necessity of assessing the risk for the population as an instrument of public health protection.

### Nucleotide sequence accession number

Information on this Whole Genome Shotgun project has been deposited at DDBJ/ENA/GenBank under the accession number JABBEB000000000 (SRA: SRR12606957). The version described in this paper is version JABBEB000000000.1.

### Supporting information

**S1 Fig. Raw gel images of PFGE performed for the 14 *L. monocytogenes* isolates.**
(PDF)

## Acknowledgments

We thank the staff at the Laboratory of Veterinary Microbiology and Laboratory of Genic Therapy for providing assistance when necessary.

## Author Contributions

**Data curation:** Bruno Stéfano Lima Dallago, Bruna Fuga.

**Investigation:** Bruno Stéfano Lima Dallago, Bruna Fuga.

**Methodology:** Emilia Fernanda Agostinho Davanzo, Rebecca Lavarini dos Santos, Virgilio Hipólito de Lemos Castro, Joana Marchesini Palma, Bruno Rocha Pribul, Margareti Medeiros, Simoneide Souza Titze de Almeida, Hayanna Maria Boaventura da Costa, Dália dos Prazeres Rodrigues, Nilton Lincopan, Simone Perecmanis.

**Supervision:** Angela Patrícia Santana.

**Writing – original draft:** Emilia Fernanda Agostinho Davanzo.

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
