## [Decision Letter · Decision Letter 0]

1 Jul 2021

PONE-D-21-17398

Molecular characterization of *Salmonella* spp. and *Listeria monocytogenes* strains from biofilms in cattle and poultry slaughterhouses located in the Federal District and State of Goiás, Brazil

PLOS ONE

Dear Dr. Agostinho Davanzo,

Thank you for submitting your manuscript to PLOS ONE. After careful consideration, we feel that it has merit but does not fully meet PLOS ONE’s publication criteria as it currently stands. Therefore, we invite you to submit a revised version of the manuscript that addresses the points raised during the review process.

There are a number of questions on methodology, data interpretation and conclusions. Please address all comments point by point. 

We look forward to receiving your revised manuscript.

Kind regards,

Iddya Karunasagar

Academic Editor

PLOS ONE

Journal Requirements:

Additional Editor Comments (if provided):

The reviewers have raised a number of important questions regarding rationale, methodology, data interpretation and conclusions. Please address all the reviewer comments point by point.

Reviewers' comments:

Reviewer's Responses to Questions

**Comments to the Author**

1. Is the manuscript technically sound, and do the data support the conclusions?

Reviewer #1: Yes

Reviewer #2: No

2. Has the statistical analysis been performed appropriately and rigorously? 

Reviewer #1: Yes

Reviewer #2: Yes

3. Have the authors made all data underlying the findings in their manuscript fully available?

Reviewer #1: Yes

Reviewer #2: Yes

4. Is the manuscript presented in an intelligible fashion and written in standard English?

Reviewer #1: No

Reviewer #2: Yes

5. Review Comments to the Author

Reviewer #1: The authors characterized two important food- borne pathogens isolated from cattle and poultry, and the study has provided interesting findings. However, the research findings are not presented in a better shape; hence, a couple of suggestions are given below for its improvement.

1. The entire manuscript has to be thoroughly re-drafted with the help of an English-speaking native researcher to ensure its improved drafting in terms of grammatical mistakes and better readability. Overall, the discussion need to be improved and comparative analysis of the assays need to be performed to draw conclusive findings.

2. Line. 133 mentions the usage of UMV broth, instead of UVM. Expand the mnemonics, while used for the first time.

3. In the antibiogram assay, mention the medium used. The quality control strains used in this assay to be mentioned.

4. The concentration of bacterial inoculum used in the in vitro biofilm assay to be mentioned in the section. How were the test isolates compared or graded for biofilm formation?

5. The expression of antibiogram (Table 3) should be modified. As there exist countable isolates, it is suggested to present an isolate- wise susceptibility pattern in the table to get a better picture. Similarly, a comparative PCR- based antibiotic susceptibility pattern should be included for better representation of data.

6. The in vitro biofilm forming assay needs to be compared with the antibiogram data as well as with the serotypes to derive findings. Overall, the discussion with regard to the biofilm forming ability and its comparison with previous assays shall be improved.

Reviewer #2: PONE-D-21-17398

Molecular characterization of Salmonella spp. and Listeria monocytogenes strains from biofilms in cattle and poultry slaughterhouses located in the Federal District and State of Goiás, Brazil.

This study aimed to detect biofilm-induced contamination points and the presence of these microorganisms in processing plants of cattle and poultry slaughterhouses, and to characterize them with respect to pathogenicity potential, biofilm formation and antimicrobial resistance.

The study could isolate 14 Listeria monocytogenes and one Salmonella enterica from poultry slaughterhouses. L. monocytogenes puslotypes 5 and 9 were more frequently isolated than others. The authors conclude that these are persistent, biofilm forming strains that might constitute constant sources of contamination of processed meat.

Comments

The number of isolates recovered in this study are limited. The relevance of whole genome sequencing of the only isolate of Salmonella to this study is not clear. Although the isolates exhibited biofilm formation, this together with the frequency of isolation of two pulsotypes does not provide strong evidence to suggest that the isolates are persistent contaminants in these facilities. Further, the sources of contamination are obscure. The manuscript has abundantly used the available literature to support their findings. However, the results available from the study are too limited to make such conclusions.

Major points

• I am not convinced with the concept biofilm-induced contamination points, which is not defined in the manuscript. Past studies have analyzed the presence of Listeria monocytogenes, Salmonella etc from processing plants by swab analysis approach. In lines 303-304, you have cited studies from the same geographical region and similar sampling points. However, this manuscript considers swab isolates form surfaces as biofilm formers. (Lines 286-288: The present study is the first investigation of L. monocytogenes biofilms in poultry and cattle slaughterhouses facilities, equipment, and utensils in the region). Again in L289-290, it is stated that “The prior findings corroborate the present results”. Corroborate what? The presence of L. monocytogenes or the biofilm L. monocytogenes?

• The persistent contaminant from biofilms across different sampling time points and locations should be clearly defined.

• Table 2: Here, the number of swabs from different collection points are different. Please explain why.

• L307-308: Here, it is stated that L. monocytogenes was detected in two samples collected on different days and this could be due to persistent, biofilm-forming strains. How close were these sampling days?. If I am correct, 5 swabs yielded equal number of isolates. How many isolates were obtained from each positive sample? It is unlikely that you got just one isolate from a sample. If you got more than one isolate, did you compare them phenotypically and genetically and then select non-repetitive isolates?.

• Again in L308-309, it is stated that “L. monocytogenes was detected in two samples collected on different days, suggesting the presence of biofilm at the site”. Did you collect samples immediately after sanitization? How do you rule out recontamination after disinfection?

• L313-314: In Berrang et al. (2010), the source of L.monocytogenes was the raw meat. According to this study, a persistent drain subtype was identical with the raw meat subtype, and this clone was recovered frequently in their sampling. However, in your study, it appears that there is no correlation between the source and the recovery of the clonal isolate from the meat chute.

• Which pulsotype was isolated frequently from meat chute? . Since you followed CDC protocol, you should be able to compare your PFGE pattern with those in PulseNet database.

• L440-442: Were L. monocytogenes resistant to chloramphenicol before the ban of the antibiotic? This is important to understand the effect of antibiotic ban on the resistance patterns.

• You have used 12oC to study biofilm formation. Does this simulate the temperature conditions of your sampling locations?

• Pulsotype 9 and 5 were more frequently isolated and probably the persistent strains as suggested in the manuscript. It is important to know if these pulsotypes exhibited higher biofilm formation compared to others which were not isolated frequently. This will strengthen your hypothesis that the persistence of these pulsotypes is due to their ability to form strong biofilms.

• The sections on antimicrobial susceptibility and biofilm formation by Salmonella Minnesota can be combined. The study could isolate a single Salmonella and the significance of this finding is very limited.

• Please avoid claiming “first time” in multiple places in the same manuscript, unless the finding is very significant.

• Materials and methods: Samples were collected from 3 poultry slaughterhouses and 2 cattle slaughterhouses. Eight visits were conducted in two facilities. If I got it correctly, it would be 8x3 =24 and 8x2-16 (40 in total). And in each visit, samples were collected from installations and the equipment. Were the same locations samples each time? If so, how many swabs from each of these? An illustration would be useful for the readers to understand your sampling strategy.

• L120: “…Biofilms were characterized..”. Biofilms can be formed on any of the above sampling locations (installations/equipment). Did you look at this possibility? Of did you select a few locations for biofilm bacteria?.

Minor points

• The abstract is too long, and you might consider shortening it.

• A Venn diagram can help understand the distribution of pulsotypes from different locations.

• I have indicated few minor corrections in the manuscript itself.

6. PLOS authors have the option to publish the peer review history of their article (what does this mean?). If published, this will include your full peer review and any attached files.

Reviewer #1: No

Reviewer #2: No

---

## [Author Response · Author response to Decision Letter 0]

12 Aug 2021

Review Comments to the Author

Reviewer #1: The authors characterized two important food- borne pathogens isolated from cattle and poultry, and the study has provided interesting findings. However, the research findings are not presented in a better shape; hence, a couple of suggestions are given below for its improvement.

1. The entire manuscript has to be thoroughly re-drafted with the help of an English-speaking native researcher to ensure its improved drafting in terms of grammatical mistakes and better readability. Overall, the discussion need to be improved and comparative analysis of the assays need to be performed to draw conclusive findings.

The manuscript was initially translated by Editage, which offered us a certificate of translation.. A new revision was made throughout the whole manuscript by the same group and we hope the readability has improved. We included the certificate of revision for consultation.

2. Line. 133 mentions the usage of UMV broth, instead of UVM. Expand the mnemonics, while used for the first time.

The mistake was corrected, and the abbreviation was expanded as indicated. (L154)

3. In the antibiogram assay, mention the medium used. The quality control strains used in this assay to be mentioned.

The medium used was mentioned as indicated (L 257/277). We did not use control strains for the antibiogram assay.

4. The concentration of bacterial inoculum used in the in vitro biofilm assay to be mentioned in the section. How were the test isolates compared or graded for biofilm formation?

The concentration of bacterial inoculum was added as indicated (L 297). The grading for biofilm formation was performed according to the classification proposed by Stepanovic et al. (2000), as stated in lines 303-305.

5. The expression of antibiogram (Table 3) should be modified. As there exist countable isolates, it is suggested to present an isolate- wise susceptibility pattern in the table to get a better picture. Similarly, a comparative PCR- based antibiotic susceptibility pattern should be included for better representation of data.

We agree with the observation and a new table comprising antimicrobial resistance and gene detection results was included, indicating as well the origin and identification of the strains (Table 4 – L437).

6. The in vitro biofilm forming assay needs to be compared with the antibiogram data as well as with the serotypes to derive findings. Overall, the discussion with regard to the biofilm forming ability and its comparison with previous assays shall be improved.

A new table (Table 6 – L 723) comprising these data was added, and the discussion was altered as well (L 708-722).

Reviewer #2: PONE-D-21-17398

Molecular characterization of Salmonella spp. and Listeria monocytogenes strains from biofilms in cattle and poultry slaughterhouses located in the Federal District and State of Goiás, Brazil.

This study aimed to detect biofilm-induced contamination points and the presence of these microorganisms in processing plants of cattle and poultry slaughterhouses, and to characterize them with respect to pathogenicity potential, biofilm formation and antimicrobial resistance.

The study could isolate 14 Listeria monocytogenes and one Salmonella enterica from poultry slaughterhouses. L. monocytogenes puslotypes 5 and 9 were more frequently isolated than others. The authors conclude that these are persistent, biofilm forming strains that might constitute constant sources of contamination of processed meat.

Comments

The number of isolates recovered in this study are limited. The relevance of whole genome sequencing of the only isolate of Salmonella to this study is not clear. Although the isolates exhibited biofilm formation, this together with the frequency of isolation of two pulsotypes does not provide strong evidence to suggest that the isolates are persistent contaminants in these facilities. Further, the sources of contamination are obscure. The manuscript has abundantly used the available literature to support their findings. However, the results available from the study are too limited to make such conclusions.

The number of isolates were products of the permissions of the industries. It was a limiting factor.

For the WGS we had an opportunity to perform this technique in association with Universidade de São Paulo, and we considered relevant to know all the resistome and virulome of the Salmonella strain detected in this study. 

All the strains were detected after rigorous process of surface sanitization in the industries, so the repeated isolation in the same place and the in vitro biofilm forming capacity must be considered. 

The purpose of this study was not to search for the sources of the contamination initially, but to detect evidence of the biofilms in industries. We’ve made some changes to the text and we hope it has improved the comprehension regarding strain persistence (L342-346).

Major points

• I am not convinced with the concept biofilm-induced contamination points, which is not defined in the manuscript. Past studies have analyzed the presence of Listeria monocytogenes, Salmonella etc from processing plants by swab analysis approach. In lines 303-304, you have cited studies from the same geographical region and similar sampling points. However, this manuscript considers swab isolates form surfaces as biofilm formers. (Lines 286-288: The present study is the first investigation of L. monocytogenes biofilms in poultry and cattle slaughterhouses facilities, equipment, and utensils in the region). Again in L289-290, it is stated that “The prior findings corroborate the present results”. Corroborate what? The presence of L. monocytogenes or the biofilm L. monocytogenes?

The biofilm characterization is defined in L140-143. A modification was made to clarify this point. We aimed to characterize these biofilm points by sampling the same points at different visits, after sanitization procedures and by performing the in vitro biofilm formation assay to draw the conclusion that they might be present in these sampling points.

In lines 293-294 we meant that the other studies corroborate the presence of the microorganism in the region. We believe the previous sentence (The present study is the first investigation of L. monocytogenes biofilms in poultry and cattle slaughterhouses facilities, equipment, and utensils in the region of the Federal District in Brazil) led to the confusing phrasing. We excluded this sentence hoping to clarify the confusion (L322-324).

• The persistent contaminant from biofilms across different sampling time points and locations should be clearly defined.

We’ve modified some sentences (L 395-399) and we hope it has improved the text. Also, the addition of the Venn diagram (Fig 2) will hopefully clarify this point.

• Table 2: Here, the number of swabs from different collection points are different. Please explain why.

The main limitation for conducting this research was the difficulty posed by the industries for participating and allowing the researchers inside their premises to perform sample collection. The different number of swabs in each visit reflects this difficulty. Sometimes we would have access to different parts of the industry, and at the next visit, that part would be out of reach. So, we aimed to collect as many points as we could in each visit, because we could not know how the next one would be. Even through this difficulty, only the points of the last visit (visit 8) could not be repeated. In all the other visits, exactly the same collection points were used, and some other points were sampled as well in order to compensate for the restriction imposed by the industries. That’s why in visits performed in abattoir A, the two last visits showed a different number of swabs than the first ones.

• L307-308: Here, it is stated that L. monocytogenes was detected in two samples collected on different days and this could be due to persistent, biofilm-forming strains. How close were these sampling days?. If I am correct, 5 swabs yielded equal number of isolates. How many isolates were obtained from each positive sample? It is unlikely that you got just one isolate from a sample. If you got more than one isolate, did you compare them phenotypically and genetically and then select non-repetitive isolates?.

As stated in line 346-348, we assured there was at least one sanitation process between sampling collection. In the case cited, the visits were one day apart, nevertheless there was cleaning and disinfection of the areas between sample collection. That is why we suggest the presence of a protective structure like biofilm that allows the microorganism to persist on the surface, even after disinfection. Yes, we had more than one isolate from each swab, but we did not perform the phenotypic and genotypic comparison between them because we had limited financial resources. We selected only one isolate per sample to perform the study. 

• Again in L308-309, it is stated that “L. monocytogenes was detected in two samples collected on different days, suggesting the presence of biofilm at the site”. Did you collect samples immediately after sanitization? How do you rule out recontamination after disinfection?

Yes, the collection was made right after sanitization, as soon as the technical manger would allow us to enter the premises. The recontamination after disinfection was ruled out by collecting samples before the beginning of a new work shift. Therefore, nothing and no one would have touched theses surfaces before sample collection.

• L313-314: In Berrang et al. (2010), the source of L. monocytogenes was the raw meat. According to this study, a persistent drain subtype was identical with the raw meat subtype, and this clone was recovered frequently in their sampling. However, in your study, it appears that there is no correlation between the source and the recovery of the clonal isolate from the meat chute.

We did not intend to detect potential sources inside the industries since we did not test meat and poultry samples, only environmental samples, but it could be our next investigation.

• Which pulsotype was isolated frequently from meat chute? . Since you followed CDC protocol, you should be able to compare your PFGE pattern with those in PulseNet database.

The PFGE analysis was conducted in partnership with the Enterobacteriaceae Laboratory / FIOCRUZ, which is the reference lab for enterobacteria in Brazil. Unfortunately, due to circumstances beyond our reach, it was not possible to access Pulsenet in order to compare our pulsotypes with the database, even though that was our initial aim. Instead, we decided to compare them amongst themselves to draw conclusions regarding pulsotype spread inside the industries and/or in the region.

• L440-442: Were L. monocytogenes resistant to chloramphenicol before the ban of the antibiotic? This is important to understand the effect of antibiotic ban on the resistance patterns.

The use of chloramphenicol was banned by the Brazilian Ministry of Agriculture in 2003. Around that time, research on the isolation and characterization of Listeria was rare and there were no reports of detection of chloramphenicol resistance in Listeria monocytogenes in the country. Since there’s a lack of research at the time, we cannot affirm that they were resistant of not before the ban. Nevertheless, we believe that the result of the sensibility is due to the long banishment period. 

• You have used 12oC to study biofilm formation. Does this simulate the temperature conditions of your sampling locations?

Yes, this is cited in line 733. We added this information in L699-700 as well.

• Pulsotype 9 and 5 were more frequently isolated and probably the persistent strains as suggested in the manuscript. It is important to know if these pulsotypes exhibited higher biofilm formation compared to others which were not isolated frequently. This will strengthen your hypothesis that the persistence of these pulsotypes is due to their ability to form strong biofilms.

The discussion regarding these results was expanded (L708-722) and a new table was added as well (Table 6, L723). 

• The sections on antimicrobial susceptibility and biofilm formation by Salmonella Minnesota can be combined. The study could isolate a single Salmonella and the significance of this finding is very limited.

In order to combine these sections, we would have to alter the whole structure of the results and discussion, that was structured by essay, and not by microorganism. We tried to make this change but the structure became confusing, so we kept the way it was. 

• Please avoid claiming “first time” in multiple places in the same manuscript, unless the finding is very significant.

Some of the places with these claims were excluded (L322-324; L363-365).

• Materials and methods: Samples were collected from 3 poultry slaughterhouses and 2 cattle slaughterhouses. Eight visits were conducted in two facilities. If I got it correctly, it would be 8x3 =24 and 8x2-16 (40 in total). And in each visit, samples were collected from installations and the equipment. Were the same locations samples each time? If so, how many swabs from each of these? An illustration would be useful for the readers to understand your sampling strategy.

There were 16 visits in total, eight in poultry slaughterhouses and eight in cattle slaughterhouses. The visits in each poultry slaughterhouse are depicted in Table 1 (Collection 1-8 in the ‘swab collection points’ line). The same locations were sampled each time, but some locations were sampled only once. It was always one swab per collection point. That happened because of restrictions that the industries would pose during visits. This resistance was the main limiting factor for performing this study.

Since the layout for each industry is very different from each other, we could not create an illustration that would help understand the sampling strategy.

• L120: “…Biofilms were characterized..”. Biofilms can be formed on any of the above sampling locations (installations/equipment). Did you look at this possibility? Of did you select a few locations for biofilm bacteria?.

We selected locations based on the available literature and on the structure and sampling availability for each industry. Points that could provide a suitable environment for microorganism adherence were chosen (crevices, corners, etc). 

Minor points

• The abstract is too long, and you might consider shortening it.

A few alterations were made to shorten the abstract.

• A Venn diagram can help understand the distribution of pulsotypes from different locations.

A Venn diagram was added (Fig2).

• I have indicated few minor corrections in the manuscript itself.

We appreciate the corrections. They were all accepted.

---

## [Decision Letter · Decision Letter 1]

25 Oct 2021

Molecular characterization of *Salmonella* spp. and *Listeria monocytogenes* strains from biofilms in cattle and poultry slaughterhouses located in the Federal District and State of Goiás, Brazil

PONE-D-21-17398R1

Dear Dr. Agostinho Davanzo,

We’re pleased to inform you that your manuscript has been judged scientifically suitable for publication and will be formally accepted for publication once it meets all outstanding technical requirements.

Kind regards,

Iddya Karunasagar

Academic Editor

PLOS ONE

Additional Editor Comments (optional):

All reviewer comments have been addressed.

Reviewers' comments:

Reviewer's Responses to Questions

**Comments to the Author**

1. If the authors have adequately addressed your comments raised in a previous round of review and you feel that this manuscript is now acceptable for publication, you may indicate that here to bypass the “Comments to the Author” section, enter your conflict of interest statement in the “Confidential to Editor” section, and submit your "Accept" recommendation.

Reviewer #1: All comments have been addressed

2. Is the manuscript technically sound, and do the data support the conclusions?

Reviewer #1: Yes

3. Has the statistical analysis been performed appropriately and rigorously? 

Reviewer #1: N/A

4. Have the authors made all data underlying the findings in their manuscript fully available?

Reviewer #1: Yes

5. Is the manuscript presented in an intelligible fashion and written in standard English?

Reviewer #1: Yes

6. Review Comments to the Author

Reviewer #1: The authors have modified the manuscript. My concerns have been addressed. I do not have further comments.

7. PLOS authors have the option to publish the peer review history of their article (what does this mean?). If published, this will include your full peer review and any attached files.

Reviewer #1: No

---

## [Editor Report · Acceptance letter]

27 Oct 2021

PONE-D-21-17398R1 

Molecular characterization of *Salmonella* spp. and *Listeria monocytogenes* strains from biofilms in cattle and poultry slaughterhouses located in the Federal District and State of Goiás, Brazil 

Dear Dr. Agostinho Davanzo:

I'm pleased to inform you that your manuscript has been deemed suitable for publication in PLOS ONE. Congratulations! Your manuscript is now with our production department. 

Kind regards, 

on behalf of

Dr. Iddya Karunasagar 

Academic Editor

PLOS ONE